# An Eco-Study for a Feasible Project: "Torun and Its Vistula Stretch—An Important Green Navigation Spot on a Blue Inland Waterway"

**Valentina-Mariana Manoiu** [1,*], **Alexandru-Ioan Craciun** [2,3], **Katarzyna Kubiak-Wójcicka** [4,*], **Marina Antonescu** [2,5] and **Bogdan Olariu** [6]

1   Department of Meteorology and Hydrology, Faculty of Geography, University of Bucharest, Blvd. Nicolae Balcescu 1, 010041 Bucharest, Romania
2   Faculty of Geography, University of Bucharest, Blvd. Nicolae Balcescu 1, 010041 Bucharest, Romania
3   IUCN European Regional Office, Bd Louis Schmidt 64, 1040 Brussels, Belgium
4   Department of Hydrology and Water Management, Faculty of Earth Sciences and Spatial Management, Nicolaus Copernicus University, Lwowska 1, 87-100 Toruń, Poland
5   National Meteorological Administration of Romania, Bucuresti-Ploiesti 97, 013686 Bucharest, Romania
6   Department of Geomorphology, Pedology and Geomatics, Faculty of Geography, University of Bucharest, Blvd. Nicolae Balcescu 1, 010041 Bucharest, Romania
*   Correspondence: valentina.manoiu@geo.unibuc.ro (V.-M.M.); kubiak@umk.pl (K.K.-W.)

**Abstract:** This paper aims to present the main trends of an eco-study for a possibly challenging future inland waterway transportation project. The study will prove if Torun and its Vistula stretch represent a viable and profitable spot on this inland waterway, and its outcomes will constitute a sound baseline that can be used for the project itself but also for many scientific, educational and economic purposes. The eco-multilayer research will comprise the following elements: hydrology and biology of the Torunian Vistula (TV) stretch, and water quality; a public opinion survey; the urban functions of the TV segment (social, educational, urban planning, aesthetical, recreational, cultural, utilitarian and economical). The conclusion is that the eco-research and the project will improve Torun City's image as a Green Urban Space, in terms of respect towards the environment, a sustainable form of transport, and attractive ways of relaxation and spending leisure time by Torun's population and visitors. The eco-study and the project will contribute to promoting the Torun region by supporting the possibility of watching beautiful landscapes (sightseeing tours) spread along the Vistula River. The project itself will have a positive impact on the Torunian economy and on the lifestyle of Torun's citizens.

**Keywords:** Torun; Vistula; eco-study; project; inland waterway transportation

## 1. Introduction

Economic development is inevitably related to human impact on the environment [1]. The scope of anthropogenic transformations of the environment, generated by the processes of urbanization and industrialization, the abusive exploitation of natural resources and the emission of greenhouse gases, caused global ecological and climatic effects that determined the conditions of life on Earth in the Anthropocene epoch [2]. The first recommendations and guidelines for global actions concerning the protection and shaping of the human environment, which should be undertaken by society in order to ensure sustainable development, were presented in the document adopted at the Earth Summit—the United Nations Conference "Environment and Development" (UNCED)—in Rio de Janeiro, in June 1992. In later years, further programs and legal acts were prepared, which proposed not only general solutions leading to the environment improvement on a global scale, but stimulated a systemic approach to local problems in connection with the global situation [3]. The intensification of actions and the implementation of some activities in Europe took

place after the publication of the next Intergovernmental Panel on Climate Change (IPCC) Report [4]. According to the authors of the IPCC report, the negative phenomena occurring in the natural environment with increasing intensity—including extreme heat, heavy rains, droughts, fires, rising sea levels, and ocean acidification—all have anthropogenic causes and will extend in many regions of the world. As a result of the increase in negative climate phenomena, in 2011 the European Union already dealt with the climatic crisis adjustment in its region, adopting the EU strategy of adaptation to climate change [5]. Cities are particularly sensitive to climate emergency, and at the same time the level of urbanization affects local climatic differences. Many cities have implemented adaptation strategies to anticipate the adverse effects of climate change and to prevent or minimize the damage [6–12]. When considering the problem of the mutual relationship between the environment and urban areas, attention should be paid to the shared connections that occur in such unique ecosystems as river valley areas. Rivers, which were the main components of ecosystems and determinants of the civilization and culture development, were reduced mainly to economic functions. Ecological research targeting sustainable urban landscapes must incorporate findings and methods from many scientific domains and directions, such as the relationship between biodiversity and ecosystem function, human role in ecosystems, landscape connectivity and resilience [13]. In addition, the structural plan should fulfill the function of equalizing individual operative elements of the city, determining the mutual relations of components, setting directions for their development, and enacting principles of cooperation between individual units, areas, and strategic points. In this case, for cities with significant culture and environmental potential [14], system solutions allowing for element coherence are represented by important structures and projects complying with green and blue infrastructure [15]. The concept of "blue-green infrastructure" emphasizes the equivalence and interdependence of green and water structures. The concept of blue-green urban networks (blue-green grids) is focused on the protection, planning and design of water structures in connection with shaping the city landscape and the system of multi-functional public spaces, but its goals also include increasing biodiversity, management of storm runoff and adaptation to climate change [16,17]. River valleys are an excellent example of connecting the blue-green infrastructure with the city functioning. The sustainable development strategy has gained its due rank in a number of Polish legal acts and political documents. The Association Agreement signed by Poland with the European Communities asserted, inter alia, that "The policy of implementing Poland's economic and social development should be guided by the principle of sustainable development. It is necessary to ensure that environmental protection requirements are incorporated into this policy from the very beginning." The blue-green infrastructure has been included in the plans for adapting to climate change in the Polish cities with more than 100,000 inhabitants. Despite its significant potential, the blue and Green Infrastructure is insufficiently researched in Poland and therefore remains scarcely used as a mechanism of counteracting the effects of climate change and adapting Polish cities to it.

The aim of this article is to present the main trends in ecological research for a potentially demanding future inland waterway transport project. The study will show whether Toruń and the Vistula are profitable points on this inland waterway, and its results will constitute a solid point of reference that can be used in the project itself, but also for many scientific, educational and economic purposes, not only local, but regional and international too.

## 2. Materials and Methods

### 2.1. Study Area

Vistula River, with its length of 1022 km, is an important symbol of Poland (Figures 1 and 2) and Polish identity [18], and it was the most navigable river in Europe four centuries ago [19,20]. It has a high economic potential (hydro-energy, navigation, water supply, and recreation), as well as important ecological functions, and it is the location of several noteworthy cities and industrial centers [18,21–24]. Vistula River and valley are important

sanctuaries for migrating and over-wintering wetland birds, 50% of which are endangered and listed in the Polish Red Data Book of Animals [25–27].

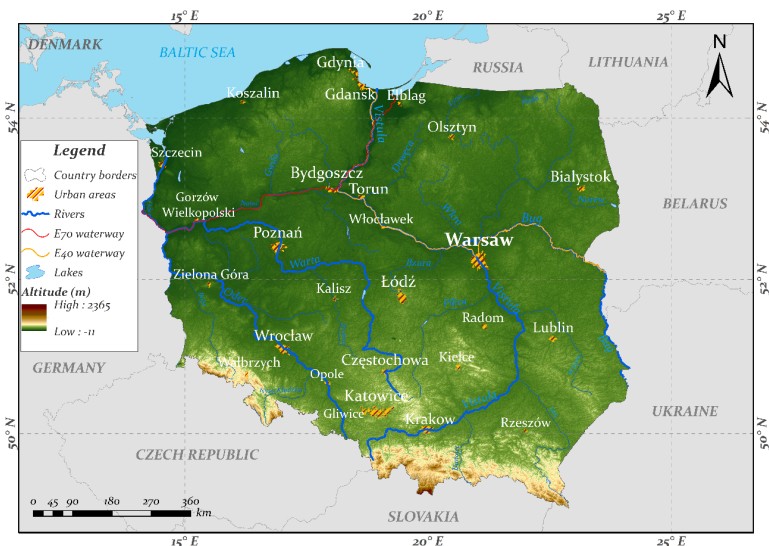

**Figure 1.** Poland and Vistula River.

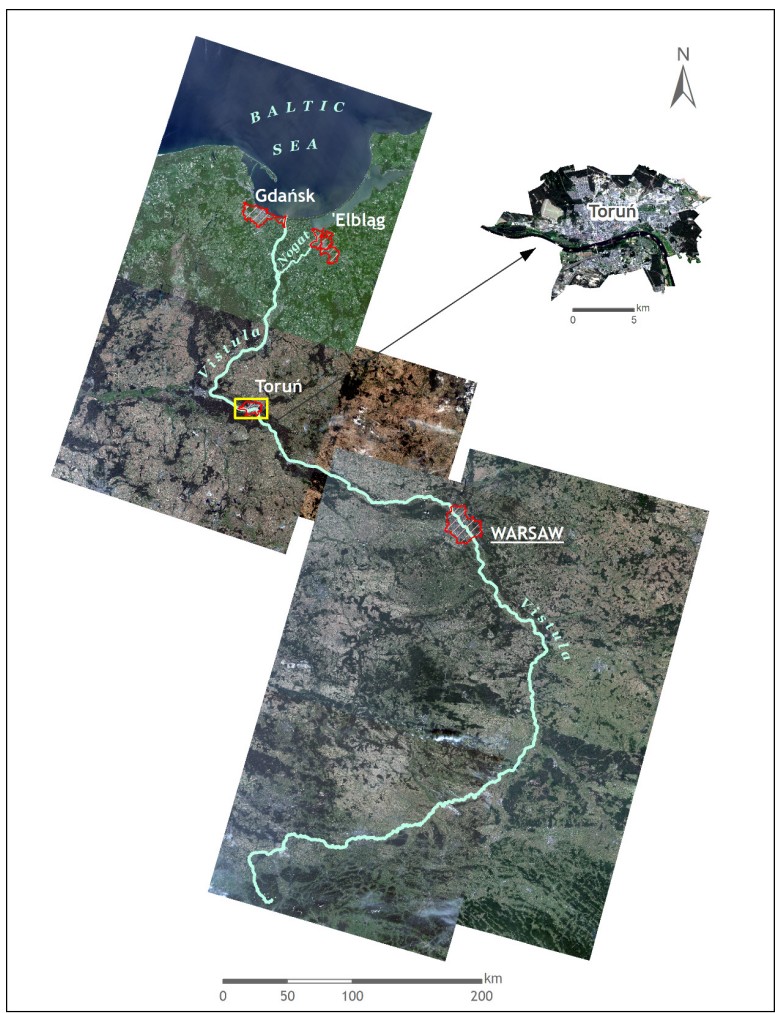

**Figure 2.** Multispectral satellite image mosaic of Landsat 9 OLI-2 from 2022 and of Landsat 8 OLI from 2019, in natural colors, 30 m resolution, of Vistula, Poland. Data source: https://earthexplorer. usgs.gov/ [28] (accessed on 11 August 2022).

Torun is located on Vistula's lower stretch, on both banks. The river runs approximately 20 km in the city and the riverbed width ranges from 355 to 440 m.

Torunian Vistula (TV) valley has a high natural value and role as an ecological corridor that provides shelter for many species [29,30], including a special protection area for birds and a natural reserve for riparian poplar and willow forest.

The population of Toruń in 2019 was 201,447 people, living in the city area totaling approximately 116 sq km. The population density was 1741 people per 1 km$^2$ [31]. The city is a unique European mixture of urban and ecological ecosystems that function in a relatively small space. Torun's historical medieval urban area has been in the UNESCO List of World Cultural Heritage since 1997 [32,33]. The city plays an important role in Kujawsko-Pomorskie Voivodeship and is its main tourist attraction.

The TV stretch (Figures 3 and 4) has many important functions, either ecological (e.g., hydrological, biological and meteorological) or urban (e.g., social, educational, urban planning, aesthetical, recreational, cultural, and utilitarian and even potentially investment-related) [29].

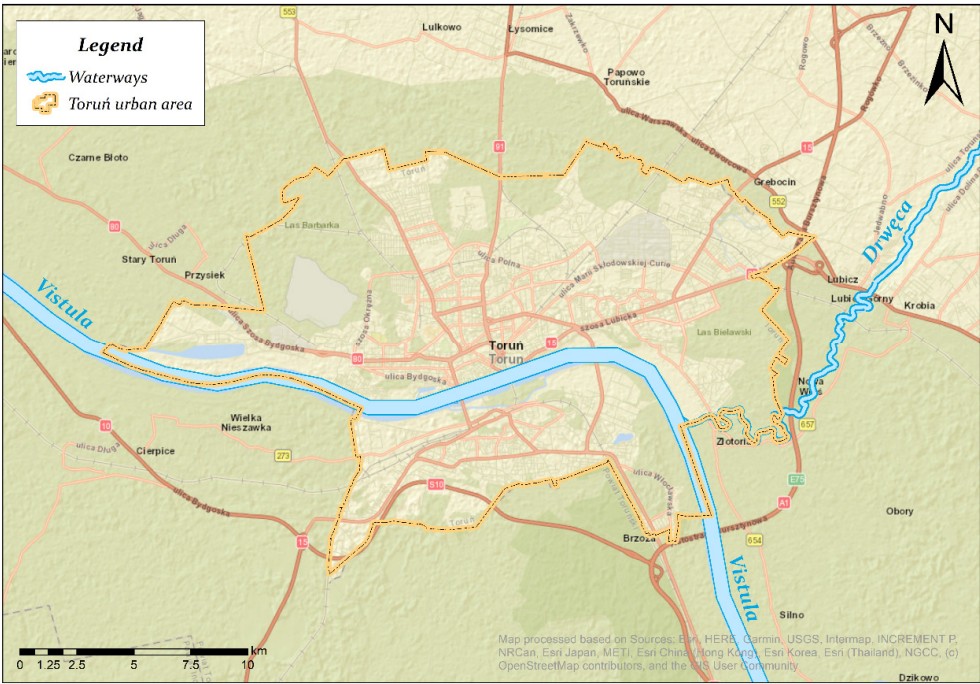

**Figure 3.** Torun and its Vistula stretch.

### 2.2. Theoretical Background

The research conducted on the Toruń section of the Vistula was carried out by many specialists in various fields. However, few studies cover an integrated view of urban space and the related elements of the geographical environment in a multidisciplinary approach. In 2017, Kubiak-Wojcicka et al. [18] applied a SWOT analysis on endogenous and exogenous factors that should impose the TV ecological and urban functions future directions. The results highlighted that Torun has a large potential for development regarding its blue urban waterway, Vistula, but we consider that there is an urgency to invest in the up-to-date and operative harbors and docks. In such a way, Torun may become *the city of the river* or *the city open to the river*, a *blue and green brand* that can bring many tourists to its space and may determine them to spend more time there and enjoy the so-called "water tourism". Our opinion is in accordance with previous studies conducted by Bolt and Jerzylo in 2013 [34] and Nones in 2021 [35] who analyzed the navigation conditions of the lower Vistula water infrastructure, focusing on river ports and the inland waterway fleet.

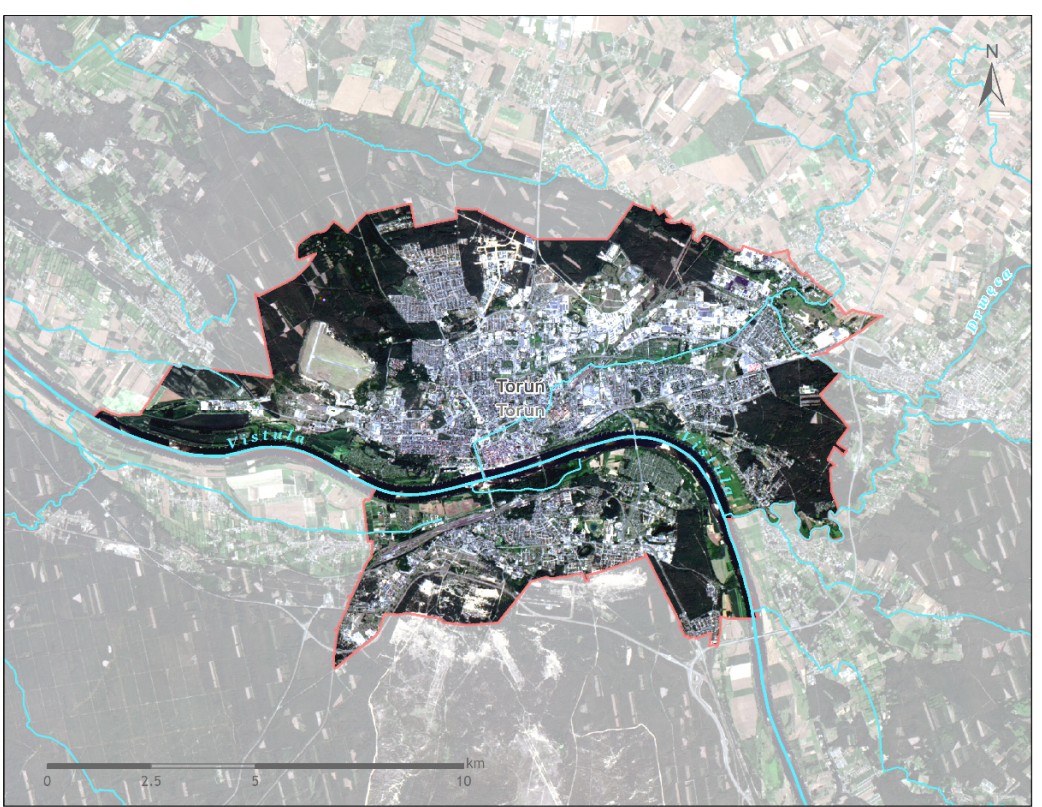

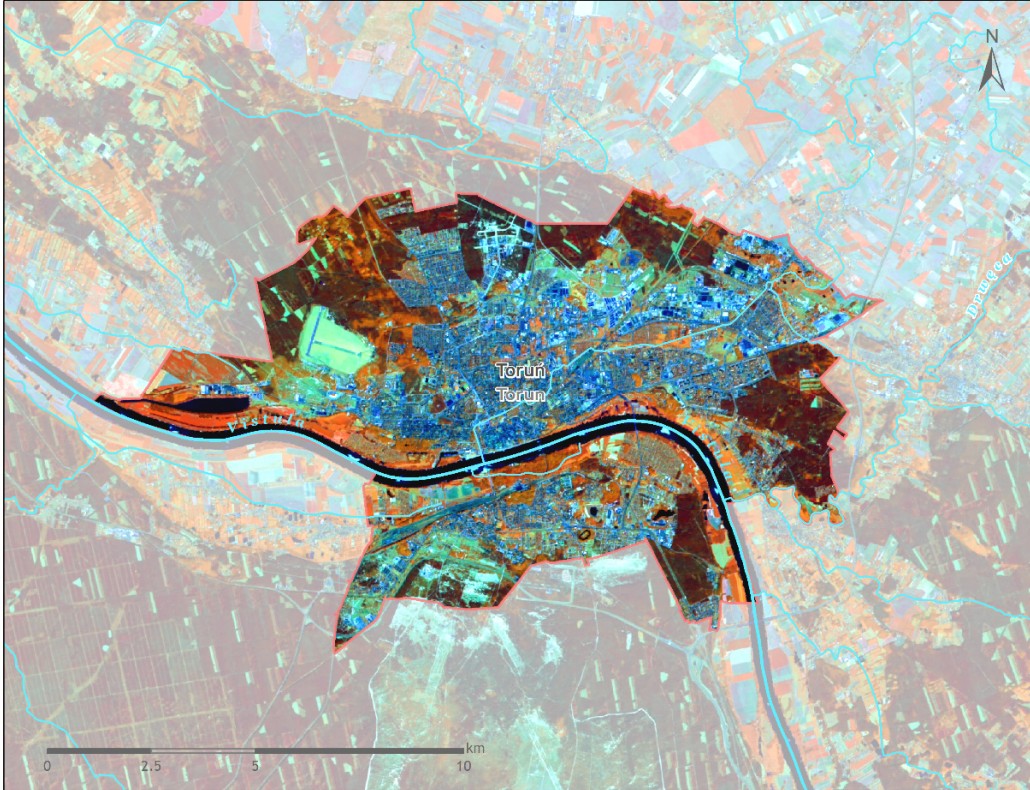

**Figure 4.** Landsat 9 OLI-2 multispectral images from July 2022, in natural colors (above) and false colors (NIR-SWIR1-RED, below; the strong hues of blue represent the urban/construction areas, while the vegetation areas are in light brown), 15 m resolution, of Torun and its Vistula stretch, Poland. Data source: https://earthexplorer.usgs.gov/ [28] (accessed on 11 August 2022).

Besides these realistic and important results, in late 2018, Anton et al. [19] sustained an old (but never put into practice) challenging idea for inland navigation: a connection between the Baltic and the Black Seas using a navigable route along the Vistula-Prut rivers waterway. The inland waterway transportation (IWT) could have a high potential in the Central and Eastern Europe countries with this interior navigation way connecting the Black and Baltic (B-B) seas, gaining a competitive advantage against other modes of transportation, more expensive and with a higher environmental impact. The idea of connecting the Baltic Sea with the Black Sea through a water road on the lower Vistula is now valid due to the construction of the canal through the Vistula Spit. The 1.3 km long canal will shorten the sea connection between the Vistula Lagoon and the Bay of Gdańsk, which will significantly improve navigation. The investment worth almost 2 billion PLN (422 million EUR) will be officially put into operation on 17 September 2022.

Having an inland waterway bridging these two important seas is not a new idea. Since 1926, the Romanian and Polish state authorities have been making "diplomatic efforts to open a new waterway connecting the Prut, Dniester, San and Vistula rivers" [19]. According to the Romanian scientists' project proposal, "navigation on the new waterway will begin at Gdansk in Poland, will continue on the river Vistula, will go on the San River connecting with the Dniester River and finally will continue on the Prut River, going to the Danube, near Galați city (Romania)". From Galați, the final port of Constanța at the Black Sea can be reached in two ways: either on the Sulina branch or on the Danube-Black Sea canal. The Gdansk-Galați inland waterway's length would be 1900 km and the distance between the Baltic Sea and Black Sea would be reduced from 7911 km to 2068 km. The Romanian scientists presented the project sustainability analysis, making reference to the economic, financial, operational, environmental, and social components.

Anton and his colleagues [19] underlined that the total external costs (excluding operating costs) for IWT were the lowest in comparison with the same costs of roads and railways. The scientists also specified that this IWT between the B-B seas could be a very environmentally-friendly means of transport for goods and people, if it is carried out within a certain framework of conditions. For this purpose, people involved in environmental protection, water management, and navigation management should clearly define the river's navigational boundaries. The researchers also recommended a SWOT analysis for developing the B-B seas IWT mid-term strategy.

Considering the systematic and accurate arguments of all these previous studies, the scientific goal of our investigation is to present the main directions of an eco-study for a possibly challenging future IWT project: *Torun and its Vistula stretch—an important spot for Green Navigation on a Blue Inland Waterway* since we hypothesized that Torun would be an important spot/port on this ambitious IWT between two seas. In the eco-study that will substantiate this achievable and profitable project, specialists in environmental protection, remote sensing, GIS, hydrology, water management, and navigation management should cooperate. The results of this study could stimulate the beginning of a very ambitious and old enterprise for both countries, Poland and Romania: an inland waterway connecting the B-B seas. The study will prove if Torun (including its Vistula stretch) is a viable and profitable spot on this inland waterway, and its outcomes will constitute a baseline that can be used for the future project itself but also for many scientific, educational and economic purposes, businesses, programs, ventures, and strategies.

The objectives of the "Torun and its Vistula stretch—an important spot for Green Navigation on a Blue Inland Waterway" project eco-study are the following:

— To boost the rethinking of a very resourceful and veteran venture for Poland and Romania: an interior navigable way connecting the B-B Seas;
— To demonstrate if Torun and its Vistula sector are a feasible and cost-effective spot on this interior navigation way, conducting an eco-multilayer-research (hydrological, biological, meteorological, environmental (water quality), social and educational research);
— To increase public awareness about the main elements of Green Navigation, using environmental websites and also Green Propaganda (short presentation, leaflets, flyers,

brochures etc.) on short cruise (45 min) vessels/ships that navigate on Torunian Vistula at present;

— To disseminate the study results through one (or more) scientific article(s);
— To inform the public about the project evolution and results on the Romanian and Polish environmental sites (e.g., Romanian Greenly Magazine website [36]).

*2.3. Methodology and Expected Results*

The eco-multilayer research will comprise the following elements of equal importance, pointed out in the Figure 5 flowchart, which will finally create a viable study:

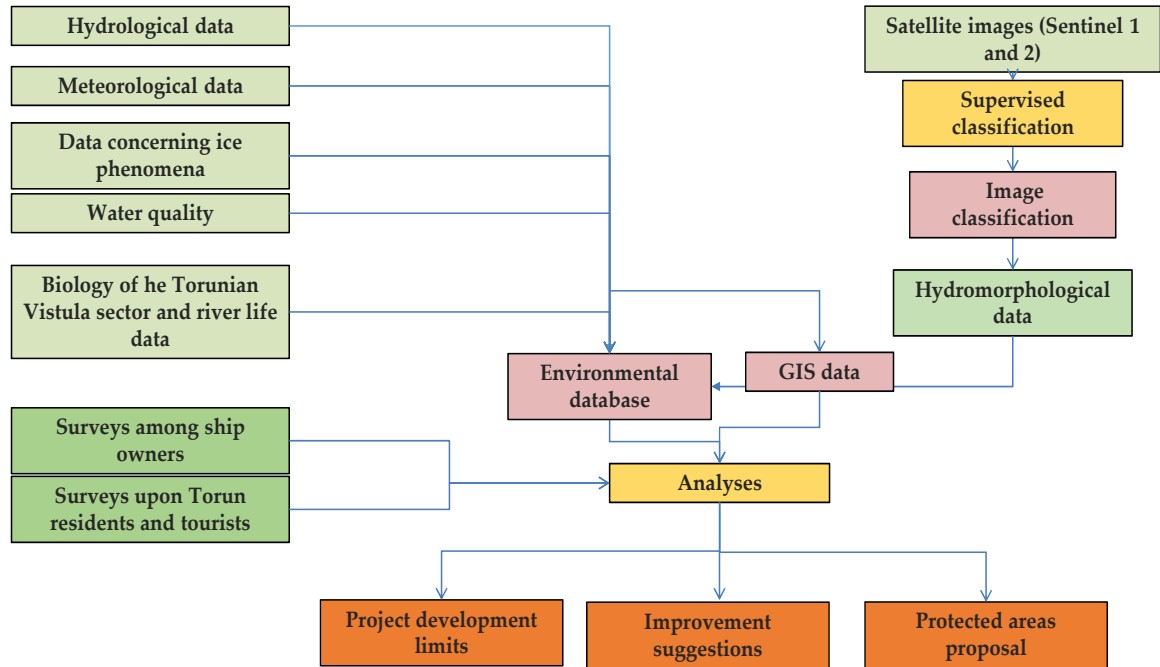

**Figure 5.** The flowchart of the research elements.

*A. Hydrology of the TV stretch, its meteorological function and the impact on navigation.* The scientists (hydrologists and meteorologists) will analyze and present the hydrological fundamentals of the Vistula stretch in Torun and its meteorological role, in connection with a potential B-B Seas IWT project.

In this respect, we must mention some important aspects.

(a) When the Vistula River basin was affected by a meteorological drought, this phenomenon influenced a larger area, comprising several European countries. The Vistula basin decreased precipitation changed the river water levels and runoff in the same direction [37,38]. For this purpose, it will be extremely important to analyze daily water levels at selected water gauges on the lower Vistula, with particular emphasis on low levels and ice phenomena. The intensity of shipping traffic mainly depends on riverbed water levels. In the case of low water levels, there are restrictions in transport traffic [39];

(b) Sentinel-1 and Sentinel-2 imagery will represent an important navigation aid for inland shipping, which requires minimum transportation depth data, offering information about the alternate sandbar movement in the Lower Vistula and the river channel geomorphology and dynamics. The alternate sandbars and their movement are an obstacle for navigation but serve as an important habitat for migrating birds [40]. The alternate sandbar speed and its significance for navigation, environment, and water management can be investigated by using the Sentinel-1 and Sentinel-2 imagery [41,42].

Besides that, Landsat 8 and 9 images are also useful for other details in connection with shipping and water and environment management. For Figures 2 and 4, we have used the data basis presented in Table 1 but also Polish GIS datasets [43], Polish GIS support [44], and Polish EnviroSolutions GIS data [45]. The satellite images have been processed with ENVI 5.3 and ArcGIS 10.3.1 ESRI Romania software packages in natural and false-colors.

**Table 1.** The data sources used for Figures 2 and 4 (United States Geological Survey data basis [28]).

| No. | Satellite Image | Sensor | Date of Image | Resolution Used | Path | Row | Access Date |
|---|---|---|---|---|---|---|---|
| 1 | Landsat 8 | OLI | 3 June 2019 | 30 m | 190 | 022 | 12 August 2022 |
| 2 | Landsat 8 | OLI | 30 June 2019 | 30 m | 187 | 025 | 11 August 2022 |
| 3 | Landsat 8 | OLI | 30 June 2019 | 30 m | 187 | 024 | 11 August 2022 |
| 4 | Landsat 8 | OLI | 25 August 2019 | 30 m | 188 | 024 | 11 August 2022 |
| 5 | Landsat 8 | OLI | 25 August 2019 | 30 m | 188 | 025 | 11 August 2022 |
| 6 | Landsat 8 | OLI | 30 August 2019 | 30 m | 189 | 023 | 12 August 2022 |
| 7 | Landsat 9 | OLI-2 | 21 July 2022 | 15/30 m | 190 | 023 | 11 August 2022 |

(c)     The operation of the Wloclawek Reservoir diminishes Lower Vistula's floods and low-water phases but by being caused by climate change the low levels become a growing threat to inland navigation [46];

(d)     Vistula's outflow is resilient to meteorological changes by causing the river's considerable hydrological inertia in connection with its basin's broad surface and varied physiography [47]. In order to explain Vistula's water resource modifications, its tributaries' water capital must be examined, together with the land use transformation and the water volume withdrawn for different economic activities. Regarding inland navigation, Vistula's water capital decreasing tendency is not acceptable and important measures in this direction should be taken;

(e)     There is no important trend in the seasonal and annual precipitation totals and flow volumes on Vistula's course, but the increasing temperatures, followed by the field evaporation growth, will enhance the summer-autumn dryness danger if the total rainfall does not change [48].

*B. Biology of the TV sector and the consequences of an IWT project on the river life.* This evaluation will highlight the importance of this specific stretch as an ecological corridor for different species [49–54] but also as a natural habitat for many plants and animals, including a special protection area for birds and a natural reserve for riparian poplar and willow forest. The impact of a future B-B seas IWT project on the biological and ecological functions of the TV segment will be thoroughly investigated, considering the fact that there is a scientific research shortage on this topic. Navigation may disturb the birds' resting and nesting, causing them to fly away and reducing the population overtime. Ship traffic can produce noise that may affect the communication between birds and their capacity to find prey underwater. Ship lights may attract, distract, or disorient birds affecting the flight to their habitat. There are tools for reducing navigation impact on birds: legal tools (laws, orders, regulations etc.) and monitoring tools. At the same time, biodiversity values can be determined using the BIO-SAFE model, which has been designed to measure biodiversity and its potential changes and to evaluate the ecotopes on the basis of legally protected species [26]. The species distribution models (SDMs) are other paradigms that can be used for the TV stretch in order to predict the biodiversity threats associated with climate change and the expansion of invasive species, the later ones being an increasing danger in the case of inland waterways [55,56].

A modern and eco-friendly approach for the TV segment and the future IWT project is to implement an ecosystem-based management (EBM) that takes into considerations both societal and ecological needs [55,56].

*C. The present TV sector water quality and its future under IWT project impact.* TV water sampling sites and quality parameters to be followed will be decided depending on the

polluting activities carried out on both banks [57–62]. We would suggest monthly analyses or, if baseline data of such water quality values already exists, they will be interrelated with a potential B-B seas IWT project and its impact on water purity.

Before choosing the water sampling points/sites/sections, all factors that affect Vistula's water quality have to be investigated. Water sampling sections must be established in places/water spots where Vistula water is homogenized, in order to collect only one water sample. We would select three water sampling sections: one should be located at the entrance of Vistula in Torun, the second one should be placed on the most polluted point of the analyzed stretch, and the third section should be set where Vistula River leaves Torun. If the time allotted to the study allows it, Vistula water quality in more sections affected by cruise navigation should be determined.

Vistula water quality can be described by only analyzing a simple combination of variables/parameters or through determination of over 100 variables/parameters. The water quality parameters chosen in this research will depend on the present and future uses of water and of polluting activities. The simplest combination of parameters that provides minimum information on water quality consists of water temperature, electrical conductivity (EC), pH, dissolved oxygen (DO), and total suspended solids (TSS). We can include more variables/parameters for determining TV stretch water quality, such as total dissolved solids (TDS), benzene, phenols, certain heavy metals etc., in connection with the supposable B-B seas IWT project. There currently are single- and multi-parameter portable meters that provide maximum flexibility for the measurements of a variety of parameter combinations, such as DO, EC, salinity, resistivity, TDS, pH, ammonia/ammonium, nitrate, chloride, temperature, etc. All of these water quality parameters can also be determined in a laboratory, using different methods, such as titration/titrimetry for DO, total nitrogen, and ammonia; atomic absorption spectrometry for the ions of heavy metals; and molecular absorption spectrometry for nitrates. For petroleum hydrocarbons there are several methods of extraction and analytical determination, each of them with its own advantages and disadvantages: infrared spectroscopy, the gravimetric method using n-hexane, gas chromatography, and ultraviolet fluorescence.

*D. Simultaneously with the first three research layers, a public opinion survey* will be conducted with Torun's population, face to face, and online. A set of relevant questions regarding all ecological and urban functions of the TV stretch and the impact of a potential B-B seas IWT project will be devised carefully. In 2015, a study of behavioral and perception geography [63], conducted by Joanna Angiel and Piotr Jan Angiel on 1900 high-school students from 11 cities along the Vistula, highlighted a large spectrum of river values recognized and perceived by students: symbolical-national, urban, natural, cultural, recreational, and sentimental. We would recommend that the Torun's entire population be targeted and that survey questions address as many functions of the river as possible.

*E. The TV segment urban functions* (social, educational, urban planning, aesthetical, recreational, cultural, utilitarian and economical ones) *and their change under an IWT project impact* will be analyzed by the scientists together with the city administration and involving the population (public opinion survey results).

*Synchronously with the last direction*, on the short cruises vessels/ships that navigate on Torunian Vistula, people will be informed about the main elements of a Green Navigation, using Green Propaganda (short presentation, leaflets, flyers, brochures etc.). In addition, different chosen environmental websites will publish articles on this same topic.

Some important attributes of Green Navigation that can be summarized either on the environmental websites or in cruise materials are the following: environmental responsibilities on board of the Green Torun cruise ships; the waste categories that can be generated on board the Green Torun ships that must be managed on various itineraries; on-board waste collection and sorting; on-board waste storage; food waste processing and disposal procedures; dangerous waste storage and packaging; medical waste; oily waste; on-board potable water and wastewater management etc.

*F. Performing the final research/study report/scientific* paper that will present the results of all research layers.

## 3. Discussions

Unfortunately, only approximately 6% of Polish waterways are suitable for modern navigation at international standards, and the remaining 94% of waterways meet regional standards [64]. The existing Polish navigable waterways have been given scarce consideration both concerning their nature and the navigation parameters, and in consequence they have a modest status and a minor function in the current Polish transportation system (0.4% of the total transported tonnage) [64]. During the last years, the Polish government has decided to reinvigorate the IWT and include it into the European system [65], ratifying the European Agreement on Main Inland Waterways of International Importance (the so-called AGN Convention) in 2017. IWT may be considered a sustainable transport practice, being one of the cheapest and most environmentally-friendly transportation types, considering the low energy consumption, low air and water pollution, and low external costs [46].

There are also many plans for developing the Lower Vistula (including its Torunian stretch), which should become a navigation link (through the E40 and E70 international European waterways represented in Figure 1) to the rest of Europe [66], along with the potential reactivation of the Lower Vistula Cascade (LVC) project [33,34]. Scientists showed that this transport project of adapting the Lower Vistula River to the International Standards of Waterway Routes will create a 177–181% profit surplus over costs (63.2%—investment costs and 36.8%—maintenance and operation costs), 41.1% of which will be transport profits connected to the environmental protection (reducing the atmosphere pollution, climate change effects, noise congestion etc.) [66]. A total of 19.6% of the profit will come from the anti-flood enactments of the project, with 0.1% from the drought anti-losses measures. The project tourism stimulation will contribute with 8.7% to the general profit, the electricity generated by the new hydroelectric power plants along the Lower Vistula will bring 6.6% to the benefits, together with 3.2% justified by the $CO_2$ emission reduction, and the protection against forest fires will provide 5.1% to the project output [66]. The final profit (15.7%) will be induced by the economic residual value.

It is relevant to specify that sometimes river regulation for an inland navigation project may involve the transfer of invasive species, water pollution, damage to the natural filter layer, reduction of the river's self-cleaning properties, threatening drinking water supplies, and diminishing the river's ecosystem services [67]. All of these aspects should be taken into consideration for the Lower Vistula IWT project development. Simultaneously, Vistula serves as a water source for agriculture and this river function should be preserved together with the inland transport role [68], contributing to the growth of plant production and the development of local food and generating farmer income and supporting the local job market etc.

At the same time, in 2017, Poland took the decision to connect the Gdansk Golf with the Vistula Lagoon through a navigable channel built in the Vistula Spit, having a lock and a small port [65]. Vistula Lagoon is befitting for tourism activities, including ferry transportation, and a low-emission navigation plan should be built [69]. Simultaneously, the Polish authorities want the Gdansk Port to be "the leading European hub in the Baltic Sea" [70], with neighboring countries, such as Czech Republic, Slovakia, Hungary, Belarus and Ukraine, being less than 800 km away from the Gdansk Port hinterland. For this purpose, many improvements and developments have been brought to the port's infrastructure and superstructure [70].

Fortunately and opportunely, Polish waterways, including Vistula, could be successfully used for touristic and recreational purposes. Gorączko and Kubiak-Wójcicka [64] concluded that "there is an increasing interest of tourists and residents in the riverside space as a place of rest and recreation" and this main investing future direction for the Polish inland navigation development is scientifically sustained by the lower transit depths required

for touristic and recreational functions, notably in dry years. Kempa and Iwanowicz [33] proposed in a pleading manner the launch of Torun Water Tram regular tourist lines, but according to our opinion, the Torunian Vistula stretch has a higher potential. Analyzing all of this information, there is no ambiguity or hesitation that the preservation of surface water naturalness and ecological function remains a priority over the economic interests [64], taking into consideration that Vistula River is a large ecological corridor used by various species, a significant part of it is under the umbrella of the Natura2000 program, and along the river there are several National Parks and Landscape Parks, many ecologists acknowledging that Vistula is the only wild river in Europe [18,23]. Undoubtedly, the presence of a river in an urban space imposes a pro-ecological development of the city and Vistula will dictate a Green Urban Economy of Torun, including the IWT.

## 4. Conclusions

Our study hypothesis is that Torun (due to its Vistula stretch) would be an important spot/port within an ambitious inland waterway transportation project located between two seas: Black and Baltic seas. If so (as it has been proved by previous investigations on Vistula and its role as a profitable interior navigable waterbody), the study will help to create new green recreational facilities and will increase opportunities for a safe, green, and operational navigation on the TV sector. The project's eco-research will maintain and perhaps even improve Torun City's image as a green urban space, in terms of ecology, respect towards the environment, a green and sustainable form of transport, and enticing options for relaxation and spending leisure time by both Torun's population and its visitors/tourists. The eco-study and the project will contribute to promoting the Torun region by ensuring and supporting the possibility of enjoying beautiful landscapes (sightseeing circuits) spread along the Vistula River. The project itself will have a positive impact on the Torunian economy and on the lifestyle of Torun's citizens. At the same time, people will be informed about the main elements of Green Navigation, using "Green propaganda" (short presentation, leaflets, flyers, brochures etc.). Additionally, environmental websites will publish articles on this same topic. The study outcomes will constitute a baseline that can be used in the future for a wide range of scientific, educational and economic purposes. We consider that the eco-study for the project "*Torun and its Vistula stretch—an important spot for Green Navigation on a Blue Inland Waterway*" will need one year to be carried out and the implementation of the actual project will necessitate a longer time and different funding sources. The project will have to adopt modern transportation technology and to be tailored to climate fluctuations and to political or even military developments.

**Author Contributions:** Conceptualization, V.-M.M., A.-I.C., K.K.-W., M.A. and B.O.; methodology, V.-M.M., A.-I.C., K.K.-W., M.A. and B.O.; software, V.-M.M., A.-I.C., M.A. and B.O.; validation, V.-M.M., A.-I.C., M.A. and B.O.; formal analysis, V.-M.M., A.-I.C., K.K.-W., M.A. and B.O.; investigation, V.-M.M., A.-I.C., M.A. and B.O.; resources, V.-M.M., A.-I.C. and K.K.-W.; data curation, V.-M.M., A.-I.C. and K.K.-W.; writing—original draft preparation, V.-M.M. and A.-I.C.; writing—review and editing, V.-M.M., A.-I.C. and K.K.-W. All authors have read and agreed to the published version of the manuscript.

**Funding:** This research received no external funding.

**Institutional Review Board Statement:** Not applicable.

**Informed Consent Statement:** Not applicable.

**Data Availability Statement:** Not applicable.

**Conflicts of Interest:** The authors declare no conflict of interest.

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
