# Peer review of "An Eco-Study for a Feasible Project: “Torun and Its Vistula Stretch—An Important Green Navigation Spot on a Blue Inland Waterway”"

_water, doi:10.3390/w14193034_

Round 1
Reviewer 1 Report
The article is well written and takes into consideration the key problems in connection with construction of international waterways suitable even for freight transport. Although the most important references are dealing with the ecological issues, a little more detailed description of the measures for the protection of ecosystems (i.e. how to mitigate the impact of increasing flow on aquatic habitats, reducing/eliminating the risk of transfer of invasive species, conservation of aquatic ecosystems in the area of river regulation, etc.) would be more convincing... I recommend some more reference literature about biodiversity conservation of aquatic ecosystems... Furthermore, the multifunctionality of these waterways should be better emphasized (tourism and economic factors are more detailed, but agriculture, drinking water supply, biological filter function of wetlands, etc. are also of key importance.
Author Response
September 14, 2022
Dear Distinguished Professors,
We would like to submit the reviewed manuscript (Brief Report) entitled “An Eco-study for a Feasible Project: “Torun and its Vistula stretch – an important Green Navigation Spot on a Blue Inland Waterway”, by Valentina-Mariana Manoiu, Alexandru-Ioan Craciun, Katarzyna Kubiak-Wójcicka, Marina Antonescu and Bogdan Olariu, for consideration by Water Journal by MDPI, special issue: “Water Quality Management and Pollution Control under the Impact of Human Actions and Climate Change”. Two of the authors are guest editors for this special issue.
In this report we present the main trends of an eco-study for a possibly challenging future inland waterway transportation project. The study will prove if Torun and its Vistula stretch represent a viable and profitable spot on this inland waterway, and its outcomes will constitute a sound baseline that can be used for the project itself but also for many scientific, educational and economic purposes.
The eco-multilayer research will comprise the following elements: hydrology and biology of the Torunian Vistula (TV) stretch, and water quality; a public opinion survey; the urban functions of the TV segment (social, educational, urban planning, aesthetical, recreational, cultural, utilitarian and economical).
The conclusion of our report is that the eco-research and the project will improve Torun City’s image as a green urban space, in terms of respect towards the environment, a sustainable form of transport, and attractive ways of relaxation and spending leisure time by Torun’s population and visitors. The eco-study and the project will contribute to promoting the Torun region by supporting the possibility of watching beautiful landscapes (sightseeing tours) spread along the Vistula River. The project itself will have a positive impact on the Torunian economy and on the lifestyle of Torun’s citizens.
We believe that the elements presented in our report will appeal to the scientists in the field and will allow the Water Journal readers to realize the significance of a possible inland waterway transportation project connecting the Black and Baltic Seas and hence the Central and Eastern Europe, our eco-study being the first step in this direction.
We would like to thank you for your important feedbacks. We added some elements regarding the biodiversity conservation of aquatic ecosystems (please see Chapter 2), and also several aspects concerning the inland waterway multifunctionality in connection with the agriculture, drinking water supply, biological filter function (please see Discussions chapter), but unfortunately there is a scientific research shortage on these topics, because it is quite a new approach.
The Introduction has been completely changed and the concept of the “blue-green infrastructure” has been analyzed in this first chapter.
The Chapter 2 has been divided in those 3 suggested subchapters (study area, theoretical background and methodology and expected results). We added the E40 and E70 waterways in the Figure 1 and we also included a flowchart of the main research elements in the subchapter 2.3.
We underline again that this is a brief report, a first stage of our research, so the results are only forecasted. We have marked the newly introduced elements with green color.
We would be very grateful to you for other suggestions and we thank you again.
We confirm that neither the manuscript nor any parts of its content are currently under consideration or published in another journal. All authors have approved the manuscript and agree with its submission to the Water Journal by MDPI, special issue: Water Quality Management and Pollution Control under the Impact of Human Actions and Climate Change.
We have no conflicts of interests to disclose and this research received no external financial support that could have influenced its outcome.
Best regards to you!
Yours respectfully,
Assoc. Prof. Dr. Valentina-Mariana Manoiu
Department of Meteorology and Hydrology
University of Bucharest, Romania

Reviewer 2 Report
Aear Author,
The discussed topic is certainly important and innovative today!
The layout of the work is unusual, possibly acceptable at Breef Retport. In Introduction, you discuss the research area - move it to chapter 2. The introduction of the work overview lacks examples relating to the approach (Green / Blue Inland Navigation definition) you want to achieve.
Chapter 2. Suggest a flowchart with your approach. What is missing is the sequence of actions and methods on the way to achieving the goal. I propose to divide this chapter into sections (Study area, Materials and data, Expected results).
Chapter 3 and 4. What can others really learn from your work? The possible "scientific" outcomes are too weakly emphasized.
Chapter 5. There are no clear results. I am not sure if the contents of this chapter are point-based. Maybe this will allow the authors to refine the results. Maybe this will allow the authors to refine the results.
Author Response

(The authors gave the same response as above.)

Round 2
Reviewer 2 Report
The authors fully complied with my recommendations. Figure 3 has no caption of the sources and the rivers are named 2x. I have no further comments.
Author Response
Assoc. Prof. Dr. Valentina-Mariana Manoiu
Department of Meteorology and Hydrology
Faculty of Geography
University of Bucharest, Romania
Bd. Nicolae Balcescu 1, 010041, Bucharest, Romania
valentina.manoiu@geo.unibuc.ro
+40744691750
September 22, 2022
Dear Distinguished Professor,
We would like to thank you again for your important feedbacks.
We specified the sources for Figure 3 (inside the map) and we deleted those two “Wisla” words. We kept the Vistula word/name in both places. We hope we understood well your request.
We would be very grateful to you for other suggestions regarding the maps or the text.
Thank you again for everything!
Best regards to you!
Yours respectfully,
Assoc. Prof. Dr. Valentina-Mariana Manoiu
Department of Meteorology and Hydrology
University of Bucharest, Romania
